# Comparison of Aging Resistance and Antimicrobial Properties of Ethylene–Norbornene Copolymer and Poly(Lactic Acid) Impregnated with Phytochemicals Embodied in Thyme (*Thymus vulgaris*) and Clove (*Syzygium aromaticum*)

**DOI:** 10.3390/ijms222313025

**Published:** 2021-12-01

**Authors:** Anna Masek, Stefan Cichosz, Małgorzata Piotrowska

**Affiliations:** 1Faculty of Chemistry, Institute of Polymer and Dye Technology, Lodz University of Technology, Stefanowskiego 16, 90-537 Lodz, Poland; Stefan.cichosz@p.lodz.pl; 2Faculty of Biotechnology and Food Sciences, Institute of Fermentation Technology and Microbiology, Lodz University of Technology, Wolczanska 71/173, 90-924 Lodz, Poland; malgorzata.piotrowska@p.lodz.pl

**Keywords:** poly(lactic acid), ethylene–norbornene copolymer, thyme, clove, solar aging

## Abstract

The effects of plant-based extracts on the solar aging and antimicrobial properties of impregnated ethylene–norbornene (EN) copolymer and poly(lactic acid) (PLA) were investigated. In this study, the impregnation yield of polyolefin, lacking in active centers capable of phytochemical bonding, and polyester, abundant in active sides, was measured. Moreover, two different extracts plentiful in phytochemicals—thyme (TE) and clove (CE)—were employed in the solvent-based impregnation process. The effect of thymol and eugenol, the two main compounds embodied in the extracts, was studied as well. Interestingly, oxidation induction times (OIT) for the impregnation of EN with thyme and clove extracts were established to be, respectively, 27.7 and 39.02 min, which are higher than for thymol (18.4 min) and eugenol (21.1 min). Therefore, an aging experiment, mimicking the full spectrum of sunlight, was carried out to investigate the resistance to common radiation of materials impregnated with antioxidative substances. As expected, the experiment revealed that the natural extracts increased the shelf-life of the polymer matrix by inhibiting the degradation processes. The aging resistance was assessed based on detected changes in the materials’ behavior and structure that were examined with Fourier-transform infrared spectroscopy, contact angle measurements, color quantification, tensile tests, and hardness investigation. Such broad results of solar aging regarding materials impregnated with thyme and clove extracts have not been reported to date. Moreover, CE was found to be the most effective modifying agent for enabling material with antimicrobial activity against *Escherichia coli* to be obtained.

## 1. Introduction

Impregnation is a simple approach that could be widely used for the introduction of varied compounds onto the surface of polymer-based products, hence contributing to the development of active packaging [1] that exhibits antimicrobial and antiaging properties or enables external stimuli-responsive behavior, among other properties. Substances adsorbed on the surface of the packaging might be gradually released into food or change color when food is stored in an inappropriate way. 

Interestingly, impregnation is one of the few methods that enables a controlled release of the compound adsorbed on the polymer surface into the food-grade product [2]. Therefore, many substances, which are able to provide new properties and could be impregnated on the surface of polymer-based material, are being deeply investigated, e.g., lavandin essential oil [3], flax oil [4], oregano essential oil [5,6], phytol [7], fat-soluble vitamins [8], and nutmeg [9]. Imperatively, these chemical compounds might also enrich the impregnated material with antimicrobial activity. 

It has been noted that among other applications, antimicrobial materials have potential application in the packaging of fresh and processed meat, fresh and smoked fish, fresh and processed fruits/vegetables, bakery products, and ready-to-eat meals [10].

Considering the huge benefits that could be provided with this class of materials, the literature is rich in information on the antimicrobial polymer-based films preparation techniques. Below, some exemplary studies are presented. First, Gaikwad et al. [11] developed functional antimicrobial linear low-density polyethylene (LLDPE) films with coatings containing different amounts of pyrogallol, a natural phenolic substance. Prepared materials demonstrated antimicrobial action against Gram-positive and Gram-negative bacteria. 

However, Villegas et al. [12] went one step further and modified a biopolymer matrix with cinnamaldehyde to create a fully compostable material. The authors employed the supercritical CO_2_ impregnation technique to incorporate cinnamaldehyde into poly(lactic acid) (PLA) and tested antimicrobial effects of modified PLA films against *Escherichia coli* and *Staphylococcus aureus*—for impregnated films, no viability was detected.

Similarly, in a study presented by Pyla et al. [13] starch-based films impregnated with tannic acid were assessed for inhibition of *E. coli* and *Listeria monocytogenes*. The modification was successful; hence, films showed antimicrobial activity in both cases. 

However, the antimicrobial behavior of various plant-originated compounds is not the only factor that encourages researchers all over the world to take advantage of these substances. Employment of phytochemicals is also favored because of their environmental neutrality [14,15,16]. Nowadays, many works are aimed not only at material recycling [17,18,19] but also at partial [20,21] or full [22,23,24,25] replacement of synthetic materials with natural analogues [26,27]. 

Current studies provide knowledge about the influence of various nature-derived components on the properties of the obtained composites [28,29]. Consequently, different natural substances [30,31,32,33] or pro-ecological compounds of reduced toxicity [32,34] are being widely incorporated into polymer-based materials, especially in the packaging industry [18,35,36].

Recently, Zhong et al. [37] presented an interesting mini-review on biodegradable polymers and green-based antimicrobial packaging materials. The authors stated that although biopolymers are environmentally friendly and are very attractive packaging materials, unfortunately, their industrial applications are still limited due to their oxygen/water vapor barrier and thermal and mechanical properties, especially when comparing costs. 

Undoubtedly, the modification of biopolymers is possible and well described in the literature [38,39,40,41]. However, this is an additional technological step that in most cases is unprofitable from the industry’s point of view. Therefore, synthetic polymers are still and probably will be the most widely used materials for packaging until the mentioned problem is solved.

Accordingly, in this study two interesting materials holding great potential in food packaging were chosen for impregnation: an easy-processable thermoplastic elastomer— namely, ethylene–norbornene copolymer (EN)—and biodegradable poly(lactic acid) (PLA). The polymer samples were impregnated with thyme/clove ethanolic extracts and thymol/eugenol solutions. 

The process performed is a scientific novelty because these substances had not been previously used to modify EN and PLA as a result of the simple process of solvent-based impregnation.

Moreover, various phytochemicals, apart from providing antimicrobial activity [42,43], might also introduce antioxidative properties [44,45,46]. This might have an influence on both the packed food and the material itself (aging resistance). Nonetheless, this second aspect is most often missed in the literature, and aging studies are rarely performed [2,12,45,47]. 

Therefore, in this study, an additional solar aging process, mimicking the full spectrum of the sunlight, was performed. In this way, the influence of adsorbed substances on the EN and PLA behavior, when subjected to radiation, was examined. The chemical composition, mechanical properties, thermal resistance, surface free energy, and color changes of the impregnated ethylene–norbornene copolymer and poly(lactic acid) samples have been reported.

## 2. Results and Discussion

### 2.1. Characterization of the Plant Extracts

Ethanol is a suitable medium for phytochemicals extraction, as most of the compounds entrapped in plants are soluble in this environment [48,49]. Moreover, it is preferred from a green chemistry point of view [50], and therefore, it was chosen for the extraction process.

Accurately weighed quantities of thyme and clove were placed in a flask and then ethanol was added. After mixing at 50 °C for 24 h, the dispersion was filtrated, and a clarified phytochemical extract was obtained (Figure 1a,b).

Ethanol extracts and additionally prepared thymol/eugenol reference solutions were analyzed with Fourier-transform infrared spectroscopy (FT-IR) to examine the presence of phytochemicals. The results of the performed investigation are presented in Figure 1d.

The chemical structures of thymol and eugenol are given in Figure 1e,f. Apart from mostly short carbon chains and possible C=C bonds (eugenol), they consist predominantly of phenyl rings substituted with hydroxyl groups, ketone moieties, and ether linkages. However, ethanolic extracts of thyme and clove obtained in this study might be rich in many different phytochemicals.

Palmieri et al. [51] noted that in thyme extracts, 21 monoterpenes, with 1 being sesquiterpene, 8 phenolic acids (e.g., gallic acid, p-OH-benzoic acid, chlorogenic acid, vanillic acid, caffeic acid, ferulic acid, rosemarinic acid), 1 phenolic monoterpene (carvacrol), and 2 flavonoids (lutein and apigenin) were identified. Nonetheless, thymol has generally been reported to be the main component of thyme extracts.

Additionally, according to de Oliveira et al. [52], clove extract features mainly eugenol, E-caryophyllene, eugenol acetate, and α-humulene, with eugenol being obtained at the highest concentrations in all essential oil fractions analyzed. A total of 19.72% of the compounds were composed of sesquiterpene hydrocarbons, 19.79% of phenolic compounds, 0.59% of oxygenated sesquiterpenes, 59.75% of oxygenated monoterpenes, and 0.06% of other compounds.

Undoubtedly, absorption bands visible in Figure 1d confirm the presence of these structures. Peak at 3318 cm^−1^ might be attributed to hydroxyl moieties [53]. Next, maxima observed at 2973 cm^−1^ and 2882 cm^−1^ could be assigned to -CH groups [2]. Additionally, peaks at 1531 cm^−1^ and 1379 cm^−1^ were visible due to the presence of phenyl aromatic rings and absorption bands between 1268 and 1087 cm^−1^, which could be related to C-O, C-O-C ester, ether, or ketone linkages [2,54]. Moreover, the peak visible at 879 cm^−1^ is probably another signal from -CH moieties [2]. Therefore, it might be concluded that the structures observed, as expected, probably consisted of phenyl rings substituted with oxygen-rich moieties and carbon chains.

### 2.2. Analysis of the Impregnation Process and Its Impact on Polymer Samples’ Properties

Impregnation was performed in a bath heated up to 50 °C, and the process lasted for 24 h. Then, polymer samples were dried to constant mass (Figure 1c). Specimens of polymeric materials were weighed before and after the modification process. In Table 1, the mass improvement in polymer samples after the impregnation is presented. 

According to the given data, the process of PLA modification was more efficient in comparison with EN treatment. Moreover, in the case of both polymeric materials, thymol-based impregnations were more successful.

The observed phenomenon might be easily explained with the nature of the impregnation processes. According to literature, the retention of the active compound is mainly achieved by means of its chemical absorption in the polymer phase (mechanism of molecular dispersion) or the entrapment of the compound in the polymer structure by means of physical forces (mechanism of incorporation or deposition). It mostly depends on polymer swelling and on the solubility of the active compound in the solvent phase [55]. Therefore, it could have been expected that EN, as an elastic material, could have swollen during the impregnation to an extent greater than rigid thermoplastic PLA, consequently providing higher yield of the process. Nevertheless, the effect was the opposite.

Deposition depends also on the affinity between an active compound and a polymer. Therefore, the higher yield of impregnation might be observed for PLA samples that eugenol, thymol, and other hydroxyl-group-rich compounds present in prepared extracts might create, e.g., hydrogen bonds with ester groups present in poly(lactic acid) chains. It should be also underlined that during the impregnation of PLA, phytochemical matrix interactions are not very strong, mostly van der Waals and hydrogen bonding.

However, during the impregnation of EN, even these forces are lacking. Ethylene–norbornene copolymer consists of long carbon chains with carbon-embodying norbornene rings, and no active sides capable of phytochemical bonding could be evidenced. Here, the kinetics of the active compound transfer depends probably mainly on the diffusion phenomenon in the amorphous phase of a polymer structure—namely, polymer zones that are more available for the diffusing compound [55]. 

For further analysis of the impregnation processes performed, the IR spectra of unmodified and modified polymer samples were determined. In Figure 2a the spectrum of modified ethylene–norbornene copolymer is visible. Some characteristic absorption bands of EN might be found: 2915 cm^−1^, 2847 cm^−1^, 1463 cm^−1^, and 718 cm^−1^. All of them are assigned to different types of vibrations of C-H moieties in the structure of aliphatic carbon chains and norbornene rings [34,56,57]. Additional peaks, e.g., 1720 cm^−1^, 1292 cm^−1^, 1172 cm^−1^, 1045 cm^−1^, and 959 cm^−1^, are visible only after the phytochemicals’ impregnation process and are generally attributed to phenyl rings substituted with either hydroxyl groups, ketone moieties, or ether linkages [2,54,58].

Similarly, in Figure 3a, FT-IR spectra of modified poly(lactic acid) specimens are presented, and some characteristic bands of PLA are observed. The peak at 1746 cm^−1^ can be attributed to C=O moieties [42]. Absorption bands at 1451 cm^−1^ and 1359 cm^−1^ are assigned to CH_2_ and CH_3_ chemical groups, respectively [59]. Maxima visible at 1180 cm^−1^ and 1080 cm^−1^ could be related to ester bonds, e.g., C-O, C-O-C linkages [60], and the absorption band at 754 cm^−1^ is probably connected with the presence of C-H moieties [57]. 

Unfortunately, the spectra of impregnated PLA samples are similar to the reference material, as the chemical moieties present in the chain of PLA and phytochemicals’ structures are alike, e.g., -OH, -C-O-C-, C=O. Consequently, the signals overlap and are troublesome to distinguish. 

However, some additional absorption bands might be noted, e.g., between 3000 and 2800 cm^−1^, which could be assigned to C-H moieties present in substituted benzene rings of phytochemicals [34,56,57]. Moreover, some variations in the region between 1500 and 1300 cm^−1^ might be connected with the presence of additional oxygen-rich chemical moieties [60,61].

Moreover, some shifts observed between the peaks identified for reference phytochemicals’ solutions and peaks visible in Figure 2a and Figure 3a suggest possible intermolecular interactions between the molecules. The absorption bands in pristine solutions/extracts were slightly shifted toward higher or lower wavenumbers than those of impregnated specimens.

Additionally, Figure 2b–f and Figure 3b–f reveal the differences in chemical composition of impregnated polymer samples after the solar aging process. They are described in the subsequent section entitled *Characterization of the aging process*.

Moving forward, to further determine the presence of natural substances impregnated on the surface and asses the thermal resistance of polymeric samples, thermogravimetric analysis (TGA) was performed. 

The results of the carried-out investigation are presented in Figure 4. It shows the weight loss of impregnated samples of EN (Figure 4a) and PLA (Figure 4b) while heated. It is clearly visible that more phytochemicals were adsorbed on the surface of the biopolymer than prior to the polymer thermal deterioration (the most visible step at TGA curve); a significant mass drop, assigned to phytochemicals desorption, might be observed only for PLA-based specimens. This corresponds with previous observations.

Moreover, according to the data revealed in Table 2, impregnated samples of poly(lactic acid) exhibited lower decomposition temperature, while EN specimens remained almost unchanged. Noteworthily, CE impregnated on EN contributed to a gentle increase in thermal resistance.

Additionally, some more tests were performed only for ethylene–norbornene copolymer-based specimens. An oxidation induction time (OIT) experiment was carried out to establish the anti-oxidative potential of phytochemicals while impregnated (Figure 5a). Prior to the measurement, the samples were prepared under the same conditions. 

Importantly, data revealed in Table 2 prove that all natural compounds adsorbed on the surface of polymers led to the increase in oxidation induction time. Generally, it might be concluded that natural extracts prepared during this research were more successful in delaying of samples’ oxidation in comparison with commercially available products—namely, thymol and eugenol. Moreover, it is the clove extract that exhibited the most positive effect on EN specimens by prolonging OIT from 6.14 to 39.02 min (onset point). The mechanism of eugenol and thymol antioxidant activity is described in the following section entitled *Characterization of the aging process*.

Moving forward, dynamic mechanical experiments were also performed for EN specimens. During the carried-out measurements three parameters were established: storage modulus (E′), loss modulus (E″), damping factor (tanδ). Storage modulus is proportional to the energy contained within the material during the deformation process, while loss modulus is proportional to the energy lost during the deformation, e.g., heating. Additionally, the ratio of the loss to the storage is the tanδ and is often called damping. It is a measure of the energy dissipation of a material. The obtained data are presented in Figure 5b–d.

It could be observed that samples varied with regard to storage modulus characteristics. E′ achieved higher values for all impregnated specimens which is well visible from the data presented in Table 3. It might be explained with the fact that during the impregnation process, partial polymer swelling and plasticization occurred. Then, the structure of polymer sample became looser, and phenol-based phytochemicals could diffuse inside the specimen structure. Benzene rings embodied in these compounds were rigid and might have promoted the material stiffening, hence increasing the storage modulus of impregnated sample. However, temperature-enhanced partial crystallization is also possible and was previously reported in literature [62].

Similarly, the above-described phenomenon might have also contributed to the generation of differences in the value of temperature at which the maximum of tanδ is detected. This temperature is assigned to the glass transition process of polymer matrix and is regarded as a glass transition temperature (T_g_). According to the data presented in Table 3, T_g_ of impregnated samples had mostly shifted towards higher temperature values which, in general, means material stiffening.

### 2.3. Characterization of the Aging Process

To examine the aging characteristics of modified materials, specimens of ethylene–norbornene copolymer and poly(lactic acid) impregnated with natural substances were subjected to solar aging lasting for, respectively, 300 h and 100 h. The differences in aging time are dictated by the different environmental resistances of the two materials [63,64].

In Figure 6a, values of hue angle for reference samples before the aging process are presented. Hue can typically be represented quantitatively by a single number, often corresponding to an angular position around a central or neutral point or axis on a color space coordinate diagram (such as a chromaticity diagram) or color wheel. 

Additionally, in Figure 6b, the quantified color change of the impregnated samples is revealed. From Figure 6a,b it is clearly visible that the samples’ color was significantly affected by the impregnation process with phytochemicals directly extracted from plants. Hence, the aging process might be successfully tracked with color analysis.

In Figure 6d,e some pictures of analyzed samples are shown. As expected, changes in the color and transparency of polymer samples subjected to solar aging could be observed with the naked eye. However, in Figure 6c, the quantified color change of the aged specimens is shown. It is clear that the color shifted for both materials while the specimens were subjected to the wide spectrum of radiation (UV, Vis, IR). Surprisingly, the sample of PLA modified with thymol solution (PLA/T), despite exhibiting almost no color change after impregnation, became brownish after the aging process.

Color changes are usually related to variations in the chemical structure of aged materials. Therefore, to analyze the chemical structure of impregnated polymer samples after the aging process, FT-IR spectra were determined. 

The results are shown in Figure 2b–f and Figure 3b–f. It is best visible on the example of EN samples that the substance impregnated on the polymer surface degraded while irradiated with solar light, leaving the polymer matrix to remain almost unchanged. It might be expected that a similar phenomenon occurred for PLA specimens. However, it was troublesome to observe and determine due to the overlapping of the signals originating from PLA and phytochemicals.

Apart from differences in visible absorption bands, some significant shifts between the peaks could be observed. The shifts might indicate some possible changes in the interactions between the molecules within the material and possible polymer chain scission [65,66,67,68]. The absorption bands in the spectra of impregnated specimens before aging were shifted toward higher or lower wavenumbers than those after the solar aging. Quantified values representing shifts between different absorption bands are presented in Table 4.

Taking into consideration data presented in Table 4 and changes observed in the spectra shown in Figure 2b–f and Figure 3b–f, it might be concluded that substances adsorbed on the surface of PLA and EN must have significantly degraded during the performed solar aging process. It is visible not only due to some imperative shifts in wavenumbers for different peaks visible in the spectra but also visible due to the absence of some absorption bands characteristic of phytochemicals.

However, the peaks assigned to pristine (not impregnated) materials remain almost unchanged, e.g., one of the peaks assigned to C-H in ethylene–norbornene copolymer structure is at 2915 cm^−1^; it shifts to 2934 cm^−1^ while impregnated with eugenol and then shifts back to 2915 cm^−1^ after the aging process. There are more examples of this phenomenon, e.g., peak 1463 cm^−1^ for EN, absorption band at 1080 cm^−1^, and at 754 cm^−1^ for PLA.

This means that first, impregnation caused some significant shifts in the spectrum due to the intermolecular interactions between the adsorbed compound and polymer matrix. Second, after the aging process, a spectrum of pristine polymer sample was regained, suggesting the degradation of the impregnated active agent and not the polymer matrix itself.

To understand the ongoing changes, the effects of the performed aging process were also investigated with contact angle measurements to determine the surface properties of the analyzed materials. Based on the obtained results, the total surface free energy (SFE) and its components were calculated (Figure 7a,b). 

It is clear that the surface free energy changed during the aging process. At the beginning, most of the polymer samples exhibited total SFE at the level of 30 mJ/m^2^. However, while samples were submitted to solar irradiation, it was raised mostly up to 40 mJ/m^2^. 

It could be explained with the development of the samples’ surfaces, e.g., creation of micropores, microcracks. Additionally, a visible increase in the polar part of SFE might be caused by both oxidation of the polymer matrix or phytochemicals adsorbed on the surface.

Moreover, mechanical properties were investigated to determine whether the impregnated materials lowered their performance during aging. Tensile strength, elongation at break, and hardness (Figure 7c,e) were determined. 

It could be observed that the properties of ethylene–norbornene copolymer remained mostly unchanged, and the impregnation process did not seem to have a significant influence on the material mechanical properties before and after the aging process. 

To the contrary, for PLA samples, an opposite situation could be observed. Not only did impregnation of phytochemicals have an influence on the mechanical properties of the polymer, but also after the aging process, material became incredibly brittle, and carrying out tension tests was impossible. Noteworthily, the hardness of the PLA samples also increased after the aging process. However, changes in mechanical properties of PLA-based specimens might be assigned not only to the oxidation processes, but also to temperature- and radiation-enhanced crystallization process [63,64].

Additionally, calculated values of carbonyl index (CI) indicate degradation of phytochemicals adsorbed on the surface of analyzed materials, especially in case of ethylene–norbornene copolymer specimens. EN has no C=O moieties in its structure and, in this case, CI is assigned only to the presence of phytochemicals. Therefore, after the aging process, almost no evidence of C=O presence, hence, phytochemicals, may be found and the CI value drops.

However, the changes in CI for PLA-based samples are more complex. The variations in carbonyl index for PLA, PLA/CE, PLA/E, PLA/TE, PLA/T samples were, respectively, +37.5%, +31.4%, +90.9%, +57.5%, −11.8%, which indicate lower oxidation during the aging process of PLA-based samples after impregnation with clove extract (CE) and thymol (T) in comparison with pristine PLA.

According to the gathered results and data presented in the literature, it might be concluded that chemical compounds contained in thyme and clove extracts or pure thymol and eugenol might exhibit an antioxidant activity under some specific conditions, as these compounds have the phenolic hydroxyl moieties that are able to intercept the free radical chain of oxidation [44,46]. 

The antioxidants are believed to donate hydrogen from the phenolic hydroxyl groups, thereby forming a stable end-product that does not initiate or propagate further oxidation [44,46]. Moreover, the phenolic group in eugenol/thymol may stabilize a radical formed on phenolic carbon with resonance structures [44]. It was also reported before that thymol was successfully used in active packaging as an antioxidant [69]. However, its behavior might be both pro- and antioxidative, depending on the conditions [46]. 

It may be concluded that the natural compounds examined in this study altered the aging properties of impregnated polymers. Moreover, plant-originated substances changed their chemical structure during the aging process, hence preventing the degradation of the polymer matrix. The results suggest that natural compounds have prohibited, to some extent, the oxidation of the polymer matrix. However, they were not able to intercept the process of, e.g., PLA crystallization during the solar aging [63,64]. Therefore, PLA-based samples hardened and became brittle.

### 2.4. Antimicrobial Activity

Finally, the antimicrobial activity of prepared polymer samples was tested against *E. coli* (Table 5). 

Previously, it was reported in the literature that clove essential oil is widely used in traditional medicine to treat fungal infections of the skin, mouth, and urinary tract [52]. Furthermore, biocomposites modified with thymol have also been studied in the literature. The antimicrobial activity of PLA films impregnated with thymol were tested against *E. coli* and *S. aureus*. Viability was not detected for either strain [70].

However, among the tested materials, the antibacterial activity against *E. coli* was found only for the ethylene–norbornene copolymer sample modified with clove extract. The remaining materials did not show any antibacterial activity. An increase in the number of bacteria comparable to the control materials was observed for them. 

This phenomenon might be related to the previously reported possible loss of biological activity of phytochemicals during traditional impregnation processes [1,71]. Here, a supercritical CO_2_ approach, enabling different operating conditions, might be a solution.

## 3. Conclusions

Ethylene–norbornene copolymer and poly(lactic acid) samples were successfully impregnated with phytochemicals via a solvent-based approach. However, impregnation with clove/thyme extracts and eugenol/thymol solutions exhibited higher yields considering the PLA modification (mass improvement from 2.0 to 7.9 wt.%) in comparison with EN treatment (mass improvement from 0.6 to 1.8 wt.%). Performed impregnation processes lowered mechanical performance of PLA and had almost no impact on the mechanical properties of EN. Moreover, the antioxidant activity was determined from differential scanning calorimetry measurement by determining oxidation induction time (OIT). In the case of the EN impregnation with thyme and clove extracts, OIT was established for, respectively, 27.7 and 39.02 min, which was higher than for thymol (18.4 min) and eugenol (21.1 min). Additionally, solar aging of the materials revealed that the natural extracts partially inhibited the oxidation of the polymer matrix, hence prolonging shelf-life of impregnated polymers and leaving the properties of the samples unchanged. Moreover, the high color changes, visible to the human eye (ΔE up to 25), observed during the aging of impregnated polymer samples, indicate a possibility of the application of the mentioned phytochemicals as color aging indicators in active packaging. Noteworthy, EN impregnated with clove extract exhibited antimicrobial activity against *E. coli*. Therefore, these investigations may broaden the application of natural extracts and solvent-based impregnation processes in the active packaging industry.

## 4. Future Perspectives and Research Ideas

At the end of this article, we would like to outline some future perspectives and research ideas that might significantly enrich the literature:investigation of the release kinetics of impregnated phytochemicals into food-grade products;tests on the properties of actual food products packed into impregnated films;combination of extraction and impregnation processes to create a simple one-pot method;trials of surface activation before the impregnation process, e.g., plasma, corona treatment;optimization of conditions of solvent-based impregnations (e.g., concentration, temperature, time) and comparison of the results with the effects of processes relying on supercritical CO_2_;further analysis of the aging processes regarding the application of plant-originated substances as color aging indicators.

## 5. Materials and Methods

### 5.1. Materials

Ethylene–norbornene copolymer (EN), trade name TOPAS Elastomer E-140, with a bulk density of 940 kg/m^3^ was purchased from TOPAS Advanced Polymers, Runheim, Germany. The melting temperature of EN is 84 °C, the Vicat softening temperature is approximately 64 °C, and MFR = 2.7 g/10 min (190 °C, 2.16 kg). In turn, poly(lactic acid) (PLA), trade name Ingeo Biopolymer PLA 4043D, containing 4.8% D-lactide was supplied by NatureWorks LLC (Minnetonka, MN, USA) in a pellet form. PLA’s density is 1.25 g/cm^3^, and its average molecular weight equals 200 kDa. The glass transition temperature of this biopolymer is between 55 and 60 °C, the melting point about 145–160 °C, and MFR = 6.0 g/10 min (ASTM D1238). The herbal raw materials used during the extraction and impregnation processes, such as thyme and cloves, were purchased at commercial grade from Natur-Vit (Pinczow, Poland). Thymol (98.5%) and eugenol (98%) were supplied by Sigma-Aldrich (Saint Louis, MO, USA). Additionally, ethanol (96%) was obtained from Chempur (Piekary Slaskie, Poland).

### 5.2. Preparation of Polymer Composite Samples

To form plate-like polymer samples, the material was placed between the plates of a steel mold heated to 160 °C and 180 °C for, respectively, EN and PLA. The parts of the mold were previously covered with Teflon foil, and the filled mold was placed between two plates of an electrically heated hydraulic press. Polymer samples were formed under a pressure of approximately 130 bar (10 min). The material was deaerated 5 times.

### 5.3. Extraction of Phytochemicals and Impregnation Process

First, 15 wt.% dispersions of herbals in ethanol were prepared (Figure 1a). Then, the extraction of phytochemicals contained in clove (CE) and thyme (TE) proceeded for 24 h at 50 °C in a flask. Such prepared extracts were filtered and clarified (Figure 1b). The polymers were then impregnated in the extracts and 15 wt.% ethanol solutions of thymol (T) and eugenol (E). For this purpose, polymer samples were weighted before the process and were then immersed in prepared four solutions and heated up to 50 °C for 24 h (Figure 1c). After this time, the samples were removed from the bath and dried for further 18 h at 30 °C until achieving constant weight. After that, they were reweighed on a laboratory balance, and the mass difference after impregnation was calculated.

### 5.4. Solar Aging Process

Solar aging process was carried out in Atlas SC 340 MHG Solar Simulator climate chamber (AMETEK Inc., Berwyn, IL, USA) equipped with a 2500 W MHG lamp. A special rare-earth halogen lamp gives a unique range of solar radiation (UV, Vis, IR). The radiation intensity equals 1200 W/m^2^ at 100% lamp power intensity. Two alternating cycles lasting 4 h were chosen: (a) 10 °C, 80% humidity, (b) 70 °C, 80% humidity. PLA was aged for 100 h, while EN was aged for 300 h. Different times of aging were justified with the results of our previous research and the lower aging resistance of PLA [63,64].

### 5.5. Methods

#### 5.5.1. Fourier-Transform Infrared Spectroscopy (FT-IR)

Fourier-transform infrared spectroscopy (FT-IR) absorbance spectra were recorded within the 4000–400 cm^−1^ range (64 scans, absorption mode). The experiment was performed with the use of a Thermo Scientific Nicolet 6700 FT-IR spectrometer equipped with diamond Smart Orbit ATR sampling accessory (Waltham, MA, USA). Based on the obtained results, carbonyl index (CI) was calculated according to Equation (1).
(1)$$CI=\frac{absorbance_{C=O\;\left(\sim 1720\;{\mathrm{cm}}^{-1}\right)}}{absorbance_{C-H\;\left(\sim 2900\;{\mathrm{cm}}^{-1}\right)}}$$

#### 5.5.2. Static Mechanical Analysis

Tensile strength (TS) and elongation at break (Eb) were determined based on the ISO-37 standard with the use of Zwick-Roell 1435 device (Ulm, Germany). Tests were carried out on a dumbbell-shaped specimen (thickness: 1 mm, width of measured fragment: 4 mm, length of measured fragment: 25 mm, total length: 75 mm, width at the ends: 12.5 mm). The samples were operated at two different crosshead speeds for EN and PLA, respectively, 500 mm/min and 50 mm/min (the initial force at the level of 0.1 N).

#### 5.5.3. Surface Free Energy (SFE)

Surface free energy was determined based on contact angle measurements (droplet of the volume approximately 2 µL) conducted for three liquids: distilled water, ethylene glycol, 1,4-diiodomethane. OCA 15EC goniometer by DataPhysics Instruments GmbH (Filderstadt, Germany) equipped with single direct dosing system (0.01–1 mL B. Braun^®^ syringe, Hassen, Germany) was employed. Surface free energy was calculated thanks to the Owens–Wendt–Rabel–Kaelble (OWRK) method [72].

#### 5.5.4. Thermogravimetric Analysis (TGA)

A Mettler Toledo TGA/DSC 1 STARe System equipped with a Gas Controller GC10 (Greifensee, Switzerland) was employed during this investigation. The measurement was divided into two steps: (1) pyrolysis argon atmosphere (temperature range: 25–600 °C, heating rate: 20 °C/min; argon flow of 60 cm^3^/min) and (2) combustion in synthetic air (temperature range: 600–900 °C, heating rate: 20 °C/min; air flow of 60 cm^3^/min). Examined samples were placed in alumina cubicles.

#### 5.5.5. Color Change

Spectrophotometer UV-VIS CM-36001 from Konica Minolta (Tokyo, Japan). Sample colors were described with the CIE-Lab system (*L*—lightness, *a*—red-green, *b*—yellow-blue). Then, color difference (Δ*E*) and hue angle (*h_ab_*) were calculated according to Equations (2) and (3), given below:(2)$$\Delta E=\sqrt{{\left(\Delta a\right)}^{2}+{\left(\Delta b\right)}^{2}+{\left(\Delta L\right)}^{2}}$$
(3)$$h_{ab}=\{\begin{array}{c} arctg\left(\frac{b}{a}\right),\;when\;a>0\wedge b>0\\ 180{}^{\circ}+arctg\left(\frac{b}{a}\right),\;when\;\left(a<0\wedge b>0\right)\vee (a<0\wedge b<0)\\ 360{}^{\circ}+arctg\left(\frac{b}{a}\right),\;when\;a>0\wedge b<0 \end{array}$$

#### 5.5.6. Hardness Tests

Shore hardness tester type C with a pressure force of 12.5 N and spring force of 806.5 cN was employed in this measurement (Zwick/Roell, Ulm, Germany). The hardness was measured according to PN-ISO 868 standard (an indenter complying with PN-93/C04206) at five places on each of the discs distant by at least 6 mm apart.

#### 5.5.7. Dynamic Mechanical Analysis (DMA)

Dynamic mechanical analysis was carried out in tension mode, employing a DMA/SDTA861e analyzer (Mettler Toledo, Greifensee, Switzerland). Investigation was carried out only for EN specimens, and it was performed in the temperature range of −60 to 60 °C, with a heating rate of 3 °C/min, a frequency of approximately 5 Hz, and a strain amplitude of 4 μm. Three parameters were determined during the analysis: storage (E′) and loss (E″) moduli and damping factor (tanδ).

#### 5.5.8. Oxidation Induction Time (OIT) Experiment

A Mettler Toledo TGA/DSC 1 STARe System equipped with Gas Controller GC10 (Mettler Toledo, Greifensee, Switzerland) was employed in this measurement. The oxidation induction time (OIT) experiment was performed at 210 °C throughout the 50 min long measurement period (synthetic air flow: 60 mL/min). Investigation was carried out only for EN samples, and the pristine EN specimen was treated with ethanol for 24 h at 50 °C before the measurement. 

#### 5.5.9. Antimicrobial Experiment

The tests were based on the Standard Test Method for Determining the Activity of Incorporated Antimicrobial Agent(s) in Polymeric or Hydrophobic Materials ASTM E2180. Test strains of *Escherichia coli* ATCC 8739 were used. The cultures were stored on slants with Merck’s (Darmstadt, Germany) TSA medium at 6 °C. The strains were activated before the experiment. On the surface of the material placed in a sterile vessel, 500 µL of the microorganism suspension in physiological saline with 0.3% agar addition was applied. Immediately after application (t = 0), the suspension was washed with 5 mL of neutralizer, and the amount of bacteria was determined by the culture method in TSA medium. The samples with the applied suspensions were incubated at 30 °C for 24 h. After this time (t = 24), the suspension was washed with 5 mL of neutralizer, and the number of microorganisms was again determined. The results are given as the number of colony forming units per 1 cm^2^ of the material surface (UFC/cm^2^). The die rates of microorganisms D for samples with antibacterial additives and control material were determined according to Equation (4):(4)$$D=\log\left(amount\;of\;microorganizms_{\;t=0\;h}\right)-\log\left(amount\;of\;microorganizms_{\;t=24\;h}\right)\;$$

## Figures and Tables

**Figure 1 ijms-22-13025-f001:**
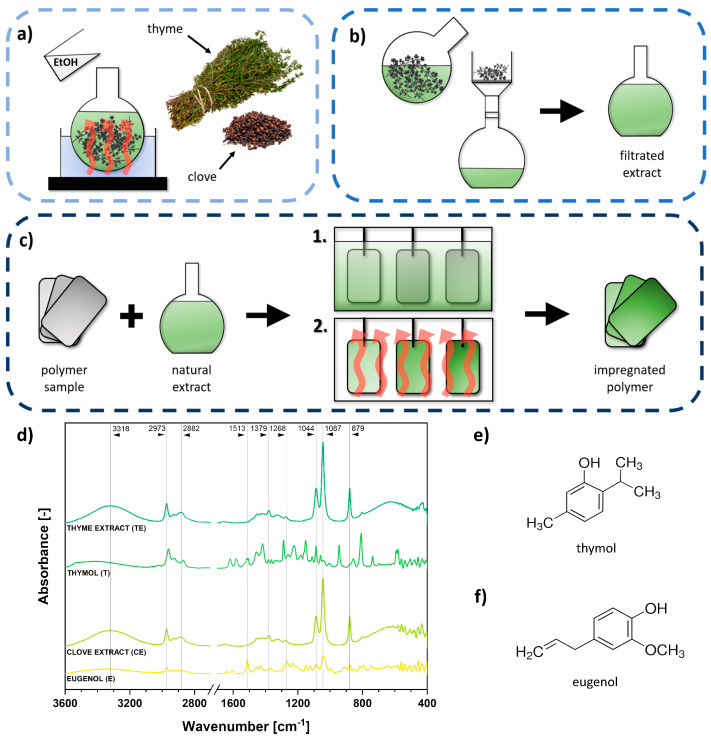
Scheme of ethylene–norbornene copolymer (EN) and poly(lactic acid) (PLA) impregnation with plant extracts: (**a**) extraction of phytochemicals in ethanol environment (50 °C, 24 h), (**b**) filtration process and purification, (**c**) proper impregnation of plastic samples with thyme/clove extracts or thymol/eugenol solutions (bath: 50 °C, 24 h; drying: 30 °C, 18 h). Solutions prepared in this study were analyzed with Fourier–transform infrared spectroscopy (**d**), and the chemical structures of two main chemical compounds present in thyme and clove, respectively, thymol (**e**) and eugenol (**f**), are shown next to the graph.

**Figure 2 ijms-22-13025-f002:**
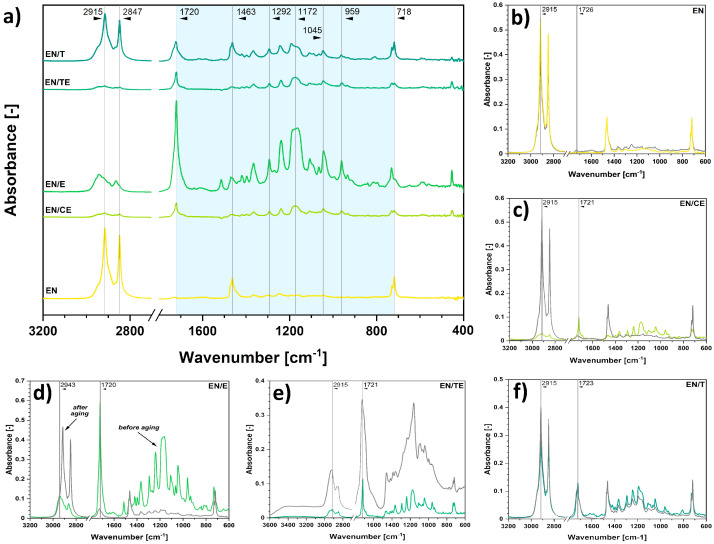
Fourier-transform infrared spectra for the impregnated samples of ethylene–norbornene (EN) copolymer (**a**) and representation of the changes observed after the aging process for: neat EN (**b**) and EN impregnated with clove extract (**c**), eugenol (**d**), thyme extract (**e**), thymol (**f**). Samples were subjected to 300 h of solar aging.

**Figure 3 ijms-22-13025-f003:**
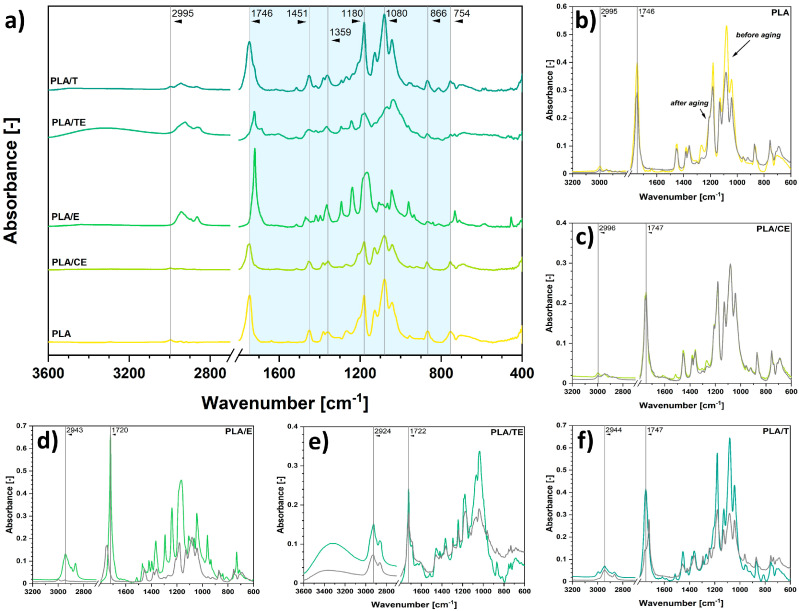
Fourier-transform infrared spectra for the impregnated samples of poly(lactic acid) (PLA) (**a**) and representation of the changes observed after the aging process for neat PLA (**b**) and PLA impregnated with clove extract (**c**), eugenol (**d**), thyme extract (**e**), thymol (**f**). Samples were subjected to 100 h of solar aging.

**Figure 4 ijms-22-13025-f004:**
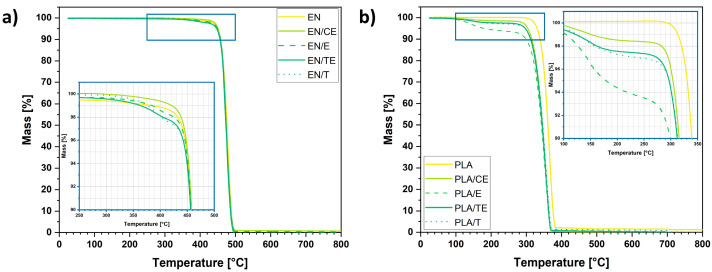
Thermogravimetric curves of ethylene–norbornene copolymer (**a**) and poly(lactic acid) (**b**) specimens.

**Figure 5 ijms-22-13025-f005:**
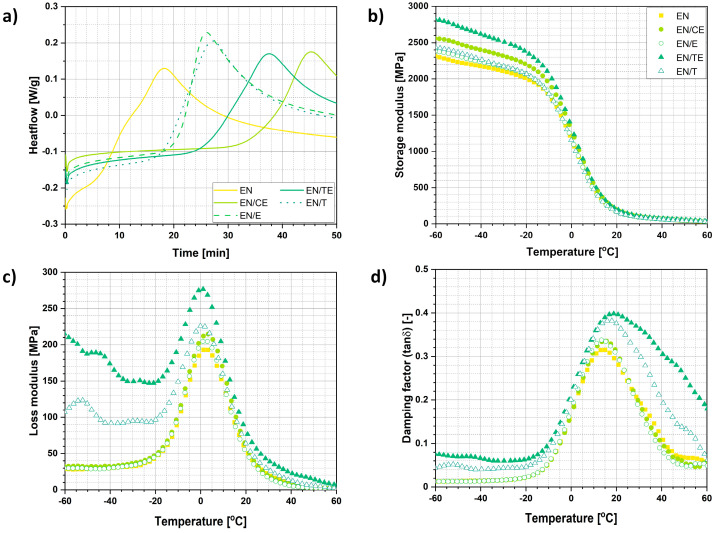
Results of additional tests performed for ethylene–norbornene copolymer specimens: oxidation induction time experiment (**a**), dynamic mechanical analysis performed from –60 to 60 °C, and the representation of changes in storage modulus (**b**), loss modulus (**c**), and damping factor (**d**).

**Figure 6 ijms-22-13025-f006:**
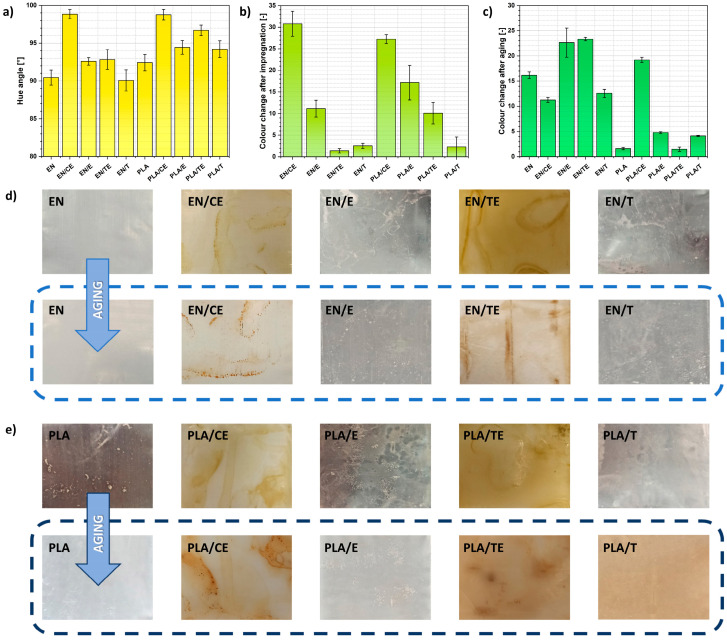
Color parameters of the polymer samples: specimens’ hue angle (**a**), color change after impregnation (**b**), color change while subjected to solar aging (**c**), as well as the pictures representing the specimens of ethylene–norbornene copolymer (**d**) and poly(lactic acid) (**e**) before and after the aging process. Photos taken on a black background.

**Figure 7 ijms-22-13025-f007:**
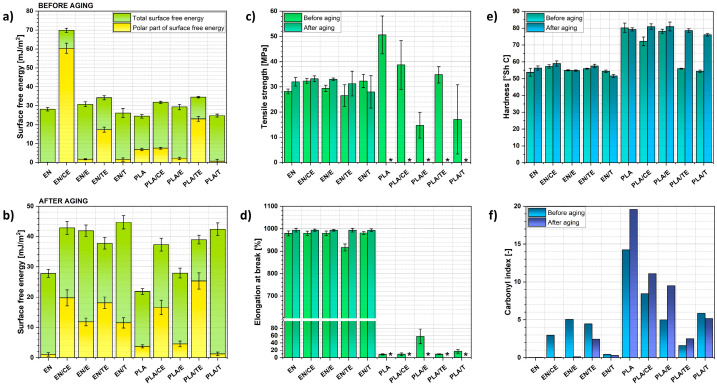
Representation of how different properties have changed during the solar aging of materials: total surface free energy and its polar component before (**a**) and after (**b**) the aging process, tensile strength (**c**), elongation at break (**d**), hardness (**e**), as well as carbonyl index calculated based on the data derived from FT-IR spectra (**f**). ∗—the measurement was unable to be performed as the polymer sample exhibited significant brittleness.

**Table 1 ijms-22-13025-t001:** Mass improvement of polymer samples after the impregnation process with different solutions.

Sample	Δm_impregnated_ (wt.%)
EN/CE	0.61
EN/E	0.74
EN/TE	1.76
EN/T	1.51
PLA/CE	2.01
PLA/E	3.06
PLA/TE	7.88
PLA/T	4.84

**Table 2 ijms-22-13025-t002:** Parameters assigned to the thermal decomposition and the oxidation process of analyzed samples; T_x_—temperature at which the mass loss of x% is detected.

Sample	Thermal Decomposition (TGA)						Oxidation Peak Parameters (DSC)		
	T_05_ (°C)	T_10_ (°C)	T_15_ (°C)	T_20_ (°C)	T_50_ (°C)	T_90_ (°C)	Onset (min)	Maximum (min)	Endset (min)
EN	449.2	456.7	461.7	464.2	475.8	487.5	6.14	18.08	31.5
EN/CE	450.8	457.5	461.7	465.0	474.2	487.5	39.02	44.65	49.83
EN/E	449.1	456.7	461.7	464.7	473.5	487.5	21.09	25.9	37.66
EN/TE	449.2	455.8	460.8	464.2	473.3	485.8	27.73	37.18	47.05
EN/T	447.5	456.7	460.8	464.2	473.5	486.7	18.41	27.08	38.58
PLA	330.0	338.3	343.3	347.5	360.8	375.8	------	------	------
PLA/CE	304.2	315.0	322.5	327.5	345.8	363.3	------	------	------
PLA/E	179.2	298.3	312.5	320.0	342.5	362.5	------	------	------
PLA/TE	296.7	311.7	320.0	325.0	345.0	363.3	------	------	------
PLA/T	296.7	3125.0	321.7	326.7	346.7	365.0	------	------	------

**Table 3 ijms-22-13025-t003:** Additional data regarding the changes in temperature properties for unimpregnated and impregnated ethylene–norbornene copolymer samples and the temperature values assigned to the maxima observed for loss modulus and damping factor; T_max E″_—temperature at which the maximum of loss modulus is detected, T_max tanδ_—temperature at which the maximum of damping factor is observed.

Sample	Storage Modulus Changes in Temperature						T_max. E″_ (°C)	T_max. tanδ_ (T_g_) (°C)
	E′_-60_ (MPa)	E′_-40_ (MPa)	E′_-20_ (MPa)	E′_0_ (MPa)	E′_20_ (MPa)	E′_40_ (MPa)		
EN	2314.7	2174.4	2009.4	1157.4	187.2	66.4	2.5	14.3
EN/CE	2560.0	2402.4	2203.4	1261.3	181.9	68.8	3.0	15.3
EN/E	2381.9	2235.3	2057.9	1166.4	167.1	63.2	1.8	13.8
EN/TE	2437.2	2269.1	2072.3	1124.2	164.2	60.1	0.8	17.2
EN/T	2830.5	2618.5	2393.0	1303.5	198.4	75.3	0.3	19.6

**Table 4 ijms-22-13025-t004:** Shifts in wavenumber observed during the solar aging process of unimpregnated and impregnated ethylene–norbornene copolymer and poly(lactic acid) specimens. Description: X_1_ → X_2_ (ΔX), where X_1_—initial wavenumber, X_2_—wavenumber assigned to the same peak after aging, ΔX—difference.

Wavenumber Shifts between the Peaks during Aging (cm^−1^)				
EN	EN/CE	EN/E	EN/TE	EN/T
2915 → 2915 (-)	2915 → 2915 (-)	2934 → 2915 (−19)	2915 → 2922 (+7)	2915 → 2915 (-)
2847 → 2847 (-)	2847 → 2847 (-)	2864 → 2847 (−17)	2847 → 2866 (+19)	2847 → 2847 (-)
	1720 → 1735 (+15)	1720 → 1724 (−4)	1721 → 1727 (+6)	1723 → 1724 (+1)
		1514 → X		
1463 → 1462 (+1)	1461 → 1462 (+1)	1470 → 1462 (−8)	1461 → 1460 (−1)	1463 → 1462 (-)
	1419 → X	1419 → X	1419 → X	
	1387 → X	1396 → X	1397 → 1388 (−9)	1366 → 1366 (-)
	1364 → 1366 (+2)	1365 → 1365 (-)	1364 → 1358 (−6)	
	1292 → X	1293 → 1294 (+1)	1292 → X	1293 → 1294 (+1)
	1239 → X	1238 → 1244 (+6)	1238 → X	1243 → 1245 (+2)
	1172 → 1154 (−18)	1162 → 1193 (+31)	1173 → 1161 (−12)	1191 → 1193 (+2)
	1108 → X	1107 → X	1107 → 1090 (−17)	1108 → 1107 (−1)
	1065 → X	1065 → X	1065 → X	
	1045 → X	1044 → 1046 (+2)	1054 → 1040 (−14)	1046 → 1047 (+1)
	959 → X	960 → X	960 → 997 (+37)	962 → X
	934 → 928 (−6)	933 → X	934 → X	
	838 → X	816 → X	840 → X	807 → X
	729 → 729 (-)	731 → 729 (−2)	729 → 729 (-)	729 → 729 (-)
718 → 718 (-)	718 → 718 (-)	718 → X	718 → 718 (-)	718 → 718 (-)
PLA	PLA/CE	PLA/E	PLA/TE	PLA/T
2995 → 2995 (-)	2996 → X	X → 2996		2994 → X
2945 → 2946 (+1)	2944 → 2944 (-)	2943 → 2947 (+4)		2944 → 2945 (+1)
2917 → X			2924 → 2927 (+3)	
		2864 → X	2863 → 2864 (+1)	2867 → 2865 (−2)
1746 → 1746 (-)	1746 → 1746 (-)	1720 → 1747 (+27)	1722 → 1722 (-)	1747 → 1722 (−25)
1451 → 1454 (+3)	1454 → 1455 (+1)	1470 → 1455 (−15)	1455 → 1456 (+1)	1452 → 1455 (+3)
1381 → 1382 (+1)	1382 → 1382 (-)	1419 → X	1419 → 1418 (−1)	1419 → 1419 (-)
1359 → 1358 (−1)	1358 → 1358 (-)	1397 → 1382 (−15)		
X → 1304	X → 1303	1365 → 1358 (−7)	1366 → 1365 (−1)	1363 → 1365 (+2)
1266 → X	1266 → 1210 (−58)	1293 → 1303 (+10)	1293 → 1293 (-)	1292 → 1293 (+1)
1180 → 1182 (+2)	1180 → 1180 (-)	1238 → 1266 (+28)	1177 → 1174 (−3)	1267 → 1242 (−25)
1127 → 1131 (+4)	1129 → 1130 (−1)	1166 → 1180 (+14)	X → 1131 (-)	1180 → 1178 (−2)
		1107 → 1129 (+22)		1127 → 1130 (+3)
1080 → 1084 (+4)	1080 → 1081 (+1)	1065 → 1081 (+16)	1066→ 1066 (-)	1080 → 1082 (+2)
1042 → 1042 (-)	1042 → 1042 (-)	1043 → 1042 (−1)	1035 → 1039 (+4)	1043 → 1042 (−1)
955 → 956 (+1)	X → 956	961 → 956 (−5)	962 → 961 (−1)	
X → 920	X → 920	X → 918		957 → 961 (+4)
866 → 871 (+5)	870 → 870 (-)	868 → 871 (+3)	X → 869	868 → 870 (+2)
754 → 754 (-)	754 → 754 (-)	732 → 755 (+23)	X → 753	754 → 754 (-)
700 → 691 (−9)	690 → 692 (+2)	X → 689	688 → 686 (−2)	695 → 688 (−7)

**Table 5 ijms-22-13025-t005:** Results of antimicrobial tests performed for unimpregnated and impregnated polymer specimens: UFC—unit-forming colony.

Sample	Amount of Microorganisms (UFC/cm^2^)		D (-)
	t = 0 h	t = 24 h	
EN	2.7 × 10^4^	4.1 × 10^5^	−1.18
EN/CE		3.0 × 10^0^	3.95
EN/E		7.3 × 10^5^	−1.43
EN/TE		5.5 × 10^5^	−1.31
EN/T		3.1 × 10^5^	−1.06
PLA	4.9 × 10^4^	3.5 × 10^5^	−0.85
PLA/CE		2.5 × 10^5^	−0.71
PLA/E		1.6 × 10^5^	−0.60
PLA/TE		1.7 × 10^6^	−1.54
PLA/T		3.5 × 10^5^	−0.85

## Data Availability

No data available while the first author was a doctoral candidate in the Interdisciplinary Doctoral School at the Lodz University of Technology, Poland.

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
