# Peer review of "Comparison of Aging Resistance and Antimicrobial Properties of Ethylene–Norbornene Copolymer and Poly(Lactic Acid) Impregnated with Phytochemicals Embodied in Thyme (Thymus vulgaris) and Clove (Syzygium aromaticum)"

_ijms, 2021, doi:10.3390/ijms222313025_

Round 1

Reviewer 1 Report

The manuscript entitled “

 Aging resistance and antimicrobial properties of ethylene-nor-2 bornene copolymer and poly(lactic acid) impregnated with 3 phytochemicals embodied in thyme (Thymus vulgaris) and 4 clove (Syzygium aromaticum)” presents a research work, But manuscript requires Major revision on basis of following points

  • I have major concern that similar kind of work has already been reported by authors then author is reporting again what is the novelty?? Masek, A.; Plota, A. Influence of a Natural Plant Antioxidant on the Ageing Process of Ethylene-norbornene Copolymer (Topas). 660 J. Mol. Sci. 2021, 22, 4018, doi:10.3390/ijms22084018.
  • Generally, the manuscript needs extensive English improvements.
  • Please incorporate some results of study in abstract section
  • In my opinion, the introduction lacks a clear structure. The overview from previous relevant research (previous related references) should be incorporated in introduction section
  • The aim of research work is not presented in clear frame.
  • The conclusion section is presented inn poor way, please explain conclusion in light of results
  • The authors have used many of their self citations which is ethically wrong, Self citations should be removed.

Author Response

Institute of Polymer and Dye Technology

Technical University of Lodz

90-924 Lodz, ul Stefanowskiego 12/16, Poland

Tel.: +48 42 631 32 23, Fax: +48 42 636 25 43

November 16, 2021

Dear Professor,

We are resubmitting our revised paper entitled Comparison of aging resistance and antimicrobial properties of ethylene-norbornene copolymer and poly(lactic acid) impregnated with phytochemicals embodied in thyme (Thymus vulgaris) and clove (Syzygium aromaticum) by Anna Masek, Stefan Cichosz, Małgorzata Piotrowska with a request to reconsider it for publication in International Journal of Molecular Sciences.

We have carefully considered the Editor and Reviewers' comments. The manuscript was revised exactly according to these comments. The list of responses to the reviewer’s comments and corrections made in the manuscript is attached.

The manuscript has not been previously published, is not currently submitted for review to any other journal, and will not be submitted elsewhere before a decision is made by this journal.

For correspondence please use the following information:

corresponding author: Anna Masek

Institute of Polymer and Dye Technology

Technical University of Lodz

90-924 Lodz, ul Stefanowskiego 12/16, Poland

Tel.: +48 42 631 32 93

Fax: +48 42 636 25 43

Yours sincerely,

Ph. D., D.Sc. Anna Masek, Associate Professor

All changes are marked with a green colour through whole manuscript.

Reviewer #1

Aging resistance and antimicrobial properties of ethylene-norbornene copolymer and poly(lactic acid) impregnated with phytochemicals embodied in thyme (Thymus vulgaris)
and clove (Syzygium aromaticum)” presents a research work, But manuscript requires Major revision on basis of following points.

The comments are listed below:

  1. I have major concern that similar kind of work has already been reported by authors then author is reporting again what is the novelty?? Masek, A.; Plota, A. Influence of a Natural Plant Antioxidant on the Ageing Process of Ethylene-norbornene Copolymer (Topas). 660 J. Mol. Sci. 2021, 22, 4018, doi:10.3390/ijms22084018.

Answer: We  are thankful for drawing our attention to his problem. However, we cannot agree with the statement that the following article debates on the same topic as the recently published article by Masek A., Plota, A. entitled Influence of a Natural Plant Antioxidant on the Ageing Process of Ethylene-norbornene Copolymer (Topas) (doi:10.3390/ijms22084018). For many years our team have been dealing with analysis of possibilities of nature-derived substances incorporation into polymer technology and its influence on weathering/aging behavior. Therefore, it is the common point of the carried out researches. The article by Masek A. and Plota A. debates on influence of hesperidin, while weathering. Contrary, the article subjected to the review is about the influence of phytochemicals embodied in clove/thyme extracts and eugenol/thymol solutions impregnation on ethylene-norbornene copolymer and poly(lactic acid) in order to assess the impregnated specimens’ resistance to solar-aging and antimicrobial activity. Additionally,  as underlined in the Abstract section, in this way, the impregnation yield of polyolefin, lacking in active centers capable of phytochemicals bonding, and polyester, abundant in active sides, could have been compared. The novelty of this research was clearly stated in the Introduction and Abstract sections:

  • Such broad results of solar aging regarding materials impregnated with thyme and clove extracts have not been reported to date.
  • The process performed is a scientific novelty because these substances had not been previously used to modify EN and PLA as a result of the simple process of sol-vent-based impregnation.

  1. Generally, the manuscript needs extensive English improvements.

Answer: The manuscript was revised concerning grammar and spelling mistakes. Language has been improved.

  1. Please incorporate some results of study in abstract section.

Answer: We are grateful for this advice. Therefore, Abstract section has been improved: The effects of plant-based extracts on the solar aging and antimicrobial properties of impregnated ethylene-norbornene (EN) copolymer and poly(lactic acid) (PLA) were investigated. In this way, the impregnation yield of polyolefin, lacking in active centers capable of phytochemicals bonding, and polyester, abundant in active sides, could have been compared. Moreover, two different extracts plentiful in phytochemicals, thyme (TE) and clove (CE), have been employed in the solvent-based impregnation process. The effect of thymol and eugenol, two main compounds embodied in the extracts, was studied as well. Interestingly, oxidation induction time (OIT) for the impregnation of EN with thyme and clove extracts were established for, respectively, 27.7 and 39.02 minutes, which is higher than for thymol (18.4 minutes) and eugenol (21.1 minutes). Therefore, the aging experi-ment, mimicking the full spectrum of the sunlight, has been carried out to investigate the resistance of materials impregnated with antioxidative substances to the common radiation. As expected, the experiment revealed that the natural extracts have increased the shelf-life of polymer matrix by inhibiting the degradation processes. The aging resistance was assessed on the basis of detected changes in materials’ behavior and structure examined with Fourier-transform infrared spectros-copy, contact angle measurements, color quantification, tensile tests and hardness investigation. Such broad results of solar aging regarding materials impregnated with thyme and clove extracts have not been reported to date. Moreover, CE was found to be the most effective modifying agent that enabled obtaining of the material with antimicrobial activity against Escherichia coli.

  1. In my opinion, the introduction lacks a clear structure. The overview from previous relevant research (previous related references) should be incorporated in introduction section.

Answer: We have revised the Introduction section. Some references to the previous relevant researches have been given. Examples are presented below:

  • However, Villegas et al. [12] have gone a one step further and modified a biopol-ymer matrix with cinnamaldehyde creating the fully compostable material. The au-thors employed supercritical CO2 impregnation technique to incorporate cinnamalde-hyde into poly(lactic acid) (PLA) and tested antimicrobial effects of modified PLA films against Escherichia coli and Staphylococcus aureus – for impregnated films no via-bility was detected.
  • Similarly, in the study presented by Pyla et al. [13] starch-based films impregnated with tannic acid were assessed for inhibition of E. coli and Listeria monocytogenes. The modification occurred to be successful, hence, films showed antimicrobial activities in both cases.
  • Recently, Zhong et al. [36] presented an interesting mini-review on biodegradable polymers and green-based antimicrobial packaging materials.

  1. The aim of research work is not presented in clear frame.

Answer: We are thankful for this comment. Nonetheless, we believe that after some improvements the article’s title in combination with Abstract and Introduction sections fully clarifies the aim of the research which is: Comparison of aging resistance and antimicrobial properties of ethylene-norbornene copolymer and poly(lactic acid) impregnated with phytochemicals embodied in thyme (Thymus vulgaris) and clove (Syzygium aromaticum). Moreover, we believe that the scope of the carried out research have been clearly presented.

  1. The conclusion section is presented inn poor way, please explain conclusion in light of results.

Answer: We tried to improve this section. Therefore, it has been altered as follows: Ethylene-norbornene copolymer and poly(lactic acid) samples were successfully impregnated with phytochemicals via a solvent-based approach. However, impregna-tion with clove/thyme extracts and eugenol/thymol solutions exhibited higher yields considering the PLA modification (mass improvement from 2.0-7.9 wt.%) in compari-son with EN treatment (mass improvement from 0.6-1.8 wt.%). Performed impregna-tion processes lowered mechanical performance of PLA and had almost no impact on mechanical properties of EN. Moreover, the antioxidant activity was determined from differential scanning calorimetry measurement by determining oxidation induction time (OIT). In case of the EN impregnation with thyme and clove extracts, OIT was es-tablished for, respectively, 27.7 and 39.02 minutes, which is higher than for thymol (18.4 minutes) and eugenol (21.1 minutes). Additionally, solar aging of the materials revealed that the natural extracts have partially inhibited the oxidation of polymer matrix, hence, prolonging shelf-life of impregnated polymers and maintaining the properties of the samples unchanged. Moreover, the high color changes, visible to a human eye (ΔE up to 25), observed during the aging of impregnated polymer samples, indicate a possibility of application of mentioned phytochemicals as color aging indi-cators in active packaging. Noteworthy, EN impregnated with clove extract occurred to exhibit the antimicrobial activity against E. coli. Therefore, these investigations may broaden the application of natural extracts and solvent-based impregnation processes in active packaging industry.

  1. The authors have used many of their self citations which is ethically wrong, Self citations should be removed.

Answer: We are terribly sorry for this mistake. We have never intended to act unethically. The reason for the mentioned amount of self-citations was the reference to the previous research results published by our group which became the starting point for this study. Self-citations were removed where possible.

Reviewer 2 Report

The abstract is hard to follow and could have been written in a more straightforward and simpler fashion. There seems to be lack of coherence in the abstract and it is challenging to comprehend it because several concepts would jump from one place to the other. It would certainly benefit this manuscript if the concept of solar aging was first introduced for readers of general background.

There are issues regarding spaces in this manuscript.

English language copyediting is needed for this paper. There are several inconsistent spelling errors noted in the manuscript.

The introduction is also hard to follow because of lack of coherence. The concept of solar aging was introduced in the abstract but it was not further explained in the introduction. Overall, it is very unclear what the manuscript intends to do and what has been done.

The rest of the manuscript is very difficult to comprehend because of the lack of coherence and organization rooting from the abstract and introduction. The introduction has been using inconsistent use of words. For example, the concept of active packaging was mentioned but it was unclear what this refers to. It was at the later point when this refers to packaging materials that was better understood.

Overall, I think this manuscript has a great potential for publication. However, it is very difficult to understand because of the lack of coherence, grammatical errors, and poor manuscript organization. It is unclear as to what the project really intends to do because of these reasons.

Author Response

Institute of Polymer and Dye Technology

Technical University of Lodz

90-924 Lodz, ul Stefanowskiego 12/16, Poland

Tel.: +48 42 631 32 23, Fax: +48 42 636 25 43

November 16, 2021

Dear Professor,

We are resubmitting our revised paper entitled Comparison of aging resistance and antimicrobial properties of ethylene-norbornene copolymer and poly(lactic acid) impregnated with phytochemicals embodied in thyme (Thymus vulgaris) and clove (Syzygium aromaticum) by Anna Masek, Stefan Cichosz, Małgorzata Piotrowska with a request to reconsider it for publication in International Journal of Molecular Sciences.

We have carefully considered the Editor and Reviewers' comments. The manuscript was revised exactly according to these comments. The list of responses to the reviewer’s comments and corrections made in the manuscript is attached.

The manuscript has not been previously published, is not currently submitted for review to any other journal, and will not be submitted elsewhere before a decision is made by this journal.

For correspondence please use the following information:

corresponding author: Anna Masek

Institute of Polymer and Dye Technology

Technical University of Lodz

90-924 Lodz, ul Stefanowskiego 12/16, Poland

Tel.: +48 42 631 32 93

Fax: +48 42 636 25 43

Yours sincerely,

Ph. D., D.Sc. Anna Masek, Associate Professor

All changes are marked with a green colour through whole manuscript.

Reviewer #2

The comments are listed below:

  1. The abstract is hard to follow and could have been written in a more straightforward and simpler fashion. There seems to be lack of coherence in the abstract and it is challenging to comprehend it because several concepts would jump from one place to the other. It would certainly benefit this manuscript if the concept of solar aging was first introduced for readers of general background.

Answer: We are thankful for this advice. Therefore, Abstract section has been improved: The effects of plant-based extracts on the solar aging and antimicrobial properties of impregnated ethylene-norbornene (EN) copolymer and poly(lactic acid) (PLA) were investigated. In this way, the impregnation yield of polyolefin, lacking in active centers capable of phytochemicals bonding, and polyester, abundant in active sides, could have been compared. Moreover, two different extracts plentiful in phytochemicals, thyme (TE) and clove (CE), have been employed in the solvent-based impregnation process. The effect of thymol and eugenol, two main compounds embodied in the extracts, was studied as well. Interestingly, oxidation induction time (OIT) for the impregnation of EN with thyme and clove extracts were established for, respectively, 27.7 and 39.02 minutes, which is higher than for thymol (18.4 minutes) and eugenol (21.1 minutes). Therefore, the aging experi-ment, mimicking the full spectrum of the sunlight, has been carried out to investigate the resistance of materials impregnated with antioxidative substances to the common radiation. As expected, the experiment revealed that the natural extracts have increased the shelf-life of polymer matrix by inhibiting the degradation processes. The aging resistance was assessed on the basis of detected changes in materials’ behavior and structure examined with Fourier-transform infrared spectros-copy, contact angle measurements, color quantification, tensile tests and hardness investigation. Such broad results of solar aging regarding materials impregnated with thyme and clove extracts have not been reported to date. Moreover, CE was found to be the most effective modifying agent that enabled obtaining of the material with antimicrobial activity against Escherichia coli.

  1. There are issues regarding spaces in this manuscript.

Answer: Unfortunately, we are not able to fully understand this comment. We suspect there is a problem with the layout of the manuscript. Nonetheless, it might be a due to the different versions of MS Word. Below we present the printscreen of initial text and figures organization:

  1. English language copyediting is needed for this paper. There are several inconsistent spelling errors noted in the manuscript.

Answer: We are grateful for this comment. The text was revised concerning grammar or spelling mistakes and the general English correctness.

  1. The introduction is also hard to follow because of lack of coherence. The concept of solar aging was introduced in the abstract but it was not further explained in the introduction. Overall, it is very unclear what the manuscript intends to do and what has been done.

Answer: We have improved the Intoduction section. We believe, now, it justifies all of the actions and presents the actual state of knowledge. Nonetheless, we cannot agree that the solar aging was not mentioned in the Intoduction section. The explanation was given in the last paragraphs: The process performed is a scientific novelty because these substances had not been previously used to modify EN and PLA as a result of the simple process of sol-vent-based impregnation.

Moreover, substances employed in this research, apart from providing antimicro-bial activity [41,42] might also introduce antioxidative properties [43–45]. This might have influence on both packed food and material itself (aging resistance). Yet, this second aspect is most often missed in literature and aging studies are barely performed [2,12,44,46].

Therefore, in this study, an additional solar ageing process, mimicking the full spectrum of the sunlight, has been performed. This way, the influence of adsorbed sub-stances on the EN and PLA behavior, while subjected to radiation, has been examined. The chemical composition, mechanical properties, thermal resistance, surface free en-ergy and color changes of the impregnated ethylene-norbornene copolymer and poly(lactic acid) samples have been reported.

  1. The rest of the manuscript is very difficult to comprehend because of the lack of coherence and organization rooting from the abstract and introduction. The introduction has been using inconsistent use of words. For example, the concept of active packaging was mentioned but it was unclear what this refers to. It was at the later point when this refers to packaging materials that was better understood.

Answer: We believe that comments no. 1 and 4 fully answers this question. The Abstract, Introduction sections have been carefully revised and improved, as well as whole manuscript.

  1. Overall, I think this manuscript has a great potential for publication. However, it is very difficult to understand because of the lack of coherence, grammatical errors, and poor manuscript organization. It is unclear as to what the project really intends to do because of these reasons.

Answer: We are grateful to the Reviewer for the kind words and we hope that after the revision everything is clearly presented and the manuscript can be easily followed

Round 2

Reviewer 1 Report

Suggested changes has been incorporated. The manuscript is accepted in present form

Author Response

Institute of Polymer and Dye Technology

Technical University of Lodz

90-924 Lodz, ul Stefanowskiego 12/16, Poland

Tel.: +48 42 631 32 23, Fax: +48 42 636 25 43

November 29, 2021

Dear Professor,

We are resubmitting our revised paper entitled Comparison of aging resistance and antimicrobial properties of ethylene-norbornene copolymer and poly(lactic acid) impregnated with phytochemicals embodied in thyme (Thymus vulgaris) and clove (Syzygium aromaticum) by Anna Masek, Stefan Cichosz, Małgorzata Piotrowska with a request to reconsider it for publication in International Journal of Molecular Sciences.

We have carefully considered the Editor and Reviewers' comments. The manuscript was revised exactly according to these comments. The list of responses to the reviewer’s comments and corrections made in the manuscript is attached.

The manuscript has not been previously published, is not currently submitted for review to any other journal, and will not be submitted elsewhere before a decision is made by this journal.

For correspondence please use the following information:

corresponding author: Anna Masek

Institute of Polymer and Dye Technology

Technical University of Lodz

90-924 Lodz, ul Stefanowskiego 12/16, Poland

Tel.: +48 42 631 32 93

Fax: +48 42 636 25 43

Yours sincerely,

Ph. D., D.Sc. Anna Masek, Associate Professor

All changes are marked with a green colour through whole manuscript.

Reviewer 2 Report

There are minor English copyediting errors.

Author Response

Institute of Polymer and Dye Technology

Technical University of Lodz

90-924 Lodz, ul Stefanowskiego 12/16, Poland

Tel.: +48 42 631 32 23, Fax: +48 42 636 25 43

November 29, 2021

Dear Professor,

We are resubmitting our revised paper entitled Comparison of aging resistance and antimicrobial properties of ethylene-norbornene copolymer and poly(lactic acid) impregnated with phytochemicals embodied in thyme (Thymus vulgaris) and clove (Syzygium aromaticum) by Anna Masek, Stefan Cichosz, Małgorzata Piotrowska with a request to reconsider it for publication in International Journal of Molecular Sciences.

We have carefully considered the Editor and Reviewers' comments. The manuscript was revised exactly according to these comments. The list of responses to the reviewer’s comments and corrections made in the manuscript is attached.

The manuscript has not been previously published, is not currently submitted for review to any other journal, and will not be submitted elsewhere before a decision is made by this journal.

For correspondence please use the following information:

corresponding author: Anna Masek

Institute of Polymer and Dye Technology

Technical University of Lodz

90-924 Lodz, ul Stefanowskiego 12/16, Poland

Tel.: +48 42 631 32 93

Fax: +48 42 636 25 43

Yours sincerely,

Ph. D., D.Sc. Anna Masek, Associate Professor

All changes are marked with a green colour through whole manuscript.

Reviewer #1

There are minor English copyediting errors.

Answer: We are thankful for drawing our attention to this problem. The linguistic correctness has been checked again. Any errors found have been corrected and the changes are marked in green in the text. Found: typos, stylistic and grammatical errors. We hope that with the changes made the manuscript is written in a legible style.